# Evaluation of Different Natural Ventilation Strategies by Monitoring the Indoor Air Quality Using CO_2_ Sensors

**DOI:** 10.3390/ijerph20186757

**Published:** 2023-09-14

**Authors:** Ana Sánchez-Fernández, Eloína Coll-Aliaga, Victoria Lerma-Arce, Edgar Lorenzo-Sáez

**Affiliations:** ITACA Research Institute, Universitat Politècnica de València, Camí de Vera s/n, 46022 Valencia, Spain

**Keywords:** CO_2_, ventilation strategies, monitoring, indoor air quality, pollution

## Abstract

Indoor air quality is a characteristic that depends on air pollutants inside a building and that can be affected by different ventilation strategies. There is strong evidence linking poor indoor air quality (IAQ) and harmful health effects, especially on vulnerable collectives, such as children in schools. Due to this concern, this work aims to provide guidance on the design of highly efficient ventilation strategies to improve the air quality of schools’ classrooms. For this, IAQ monitoring has been carried out in eight educational in real conditions centres using CO_2_ concentration as an IAQ indicator. Variables such as the presence of students and their number, activity developed in the classroom and ventilation strategy used together with break time duration have been also recorded to analyse their influence on CO_2_ concentration levels. Concluding results have allowed us to determine the maximum number of students allowed in a closed room to maintain CO_2_ levels at normal concentrations and the time needed to reduce these CO_2_ levels depending on the ventilation strategy adopted. Moreover, it has been discussed how surrounding school conditions (pollution or noise) and the building isolation are impacting the final IAQ in the classrooms studied.

## 1. Introduction

Indoor air quality (IAQ), together with thermal and hygrometric comfort, lighting, and acoustic quality, is an indoor environmental quality (IEQ) factor describing the indoor environment of a closed space [1]. There is incontestable evidence that links poor IAQ and harmful health effects inducing respiratory and cardiopulmonary pathologies [2].

Carbon dioxide (CO_2_) concentration is an adequate indicator of human body odour acceptability and environmental comfort [3,4]. Moreover, CO_2_ can be used as an indicator of indoor air quality [5] and it has been used as a tracer gas to evaluate ventilation techniques [3]. Even though CO_2_ concentration does not reflect the concentration of other air pollutants [3], it can help evaluate the IAQ in spaces without other major sources of pollutants or very similar conditions.

The effects on human health of exposure to CO_2_ concentration have been widely studied under different parameters and different durations of exposure. These effects can be classified as physical, psychological, and physiological and are often interrelated [6].

On the one hand, several studies have reported that the physical effects of exposure to higher CO_2_ concentration are respiratory problems and Sick Building Syndrome (SBS) [7,8]. Daisey et al. (2003) [9] verified symptoms of dizziness or fatigue at high CO_2_ concentrations by analysing the status of occupants at CO_2_ concentration levels between 1500 ppm and 4000 ppm. Nevertheless, respiratory symptoms, including breathing difficulties, sneezing, cough, and shortness of breath, were reported to be higher at a CO_2_ concentration of 800 ppm, and the American Society of Heating, Refrigerating, and Air- Conditioning Engineers (ASHRAE) threshold standard limit of CO_2_ was 1000 ppm [10], even though they have eliminated this absolute threshold in their last recommendations.

On the other hand, the psychological effect of exposure to CO_2_ was established by the reduced decision-making performance of the occupants. The physiological measurements conducted at elevated CO_2_ concentration with bio effluents reported a higher CO_2_ concentration in tissues, variation in heart rate, and increased blood circulation [11,12,13]. Another way to measure the psychological response due to indoor air pollutants is by indoor air quality (IAQ) satisfaction, a verbal scale determined by a questionnaire survey. Thus, Tsai et al. (2012) [10] surveyed the IAQ satisfaction of 396 workstation offices and showed that 70–80% of the occupants were satisfied with CO_2_ concentrations of 1020 ppm to 1080 ppm.

Students spend a significant part of their school time in classrooms with high occupancy ratios and, in most cases, poorly ventilated. Daisey et al. [9] reviewed over 300 articles that examined a wide range of CO_2_ concentrations, as indicators of poor ventilation in schools and verified that occupants were negatively affected by a CO_2_ concentration of over 1500 ppm. Moreover, age is also a factor in vulnerability to a low IAQ, as children and elders are susceptible [14], so, students can be considered a vulnerable group to poor indoor air quality.

Exposure to CO_2_ concentrations greater than 1000 ppm showed reduced attendance in schools [15] and increased symptoms of headache, fatigue, and difficulty in concentration [16]. In addition, in the case of children students, the exposure to contaminants is far more critical as they inhale more air per unit of body weight and present higher resting metabolic rates compared to adults [17]. These larger specific doses involve a lesser ability to deal with toxic chemicals [18].

This paper describes a study carried out on IAQ in several schools in Valencia, Spain, in 2019 and its main objective is to develop a replicable tool to improve indoor air quality by reducing the CO_2_ concentration through natural ventilation techniques in classrooms, assuming low CO_2_ concentrations in outside air and considering classroom occupants the only possible pollutant source.

## 2. Materials and Methods

Several factors that affect the IAQ were recorded during the monitored period: age of people in the room, ratio volume per person, the type of activity happening inside the room and whether there was any kind of mechanical ventilation.

The study was carried out in classrooms without important sources of pollutants, except occupants in the room and outdoors pollutants. During the measurement, the outdoor CO_2_ concentration was recorded to study its variations with the indoor results.

### 2.1. Characterisation of the Study Cases

Each study case is a classroom monitored during a school day. Study cases were recorded in January 2019 and November 2019 with similar weather conditions.

One educational centre was visited per day, and during that day several classrooms were recorded in the same centre. Therefore, there are centres with multiple study cases.

#### 2.1.1. Surroundings of the Educational Centres

Understanding the location and surroundings of the centres is essential when analysing the results since the outside air affects inside air quality. Ventilation involves the exchange of air with the outside, filtered or not, and depending on how hermetic the building is, there might be a constant air exchange. In addition, louder surroundings, high pollution or bad weather can cause a reduction in ventilation frequency when the only ventilation method is natural ventilation by windows opening. In this study, all the measurements were conducted during winter.

This study was developed by monitoring eight different schools using different scenarios regarding classroom ventilation.

During January 2019, five educational centres, three located in urban and two rural areas, were monitored with the following characteristics:

(a)Urban schools:CEIP Humanista Mariner: This school is in València, a city of 795.736 inhabitants in 2019 [19]. The educational centre is located in an urban area with some green spaces in the building surroundings. Three classrooms were selected for the study.Escuela Infantil Febes: This is a children’s school next to the CEIP Humanista Mariner Centre. Thus, it is located in an urban area close to green zones in the city of València. One classroom was selected for the study.CEIP San Juan de la Ribera: This primary school is also in València. It is in the city centre in a dense urban area without green zones. However, it is an area with less traffic due to the smaller streets. Two classrooms participated in the study.(b)Rural schools:CEIP Fabián y Fuero: It is located in Villar del Arzobispo, a small town of 3.554 inhabitants in 2019 [19]. The school is in the suburbs of the town, near a small industrial park and an open mine. Therefore, there are high dust concentrations and pollution in the air. The measurements were taken in two classrooms and an English laboratory.IES La Serranía: This high school is also in Villar del Arzobispo, with the same location and surroundings characteristics. One classroom, with the oldest children, participated in the study.

In November 2019, three urban educational centres were visited:CEIP Ciudad de Bolonia: located in València, in the same neighbourhood as Humanista Mariner and Febes, an urban area with nearby small and medium green zones. One classroom of this school participated in the monitoring.CEIP 8 de Març: This centre is located in the same neighbourhood but it is right in front of a big park. Only one classgroup participated in the study, but the measurements were taken in two classrooms, their daily classroom and the music classroom, so two study cases were obtained.CEIP Jaume I: This primary school, located in València, is in a denser urbanised neighbourhood with more traffic but next to a big park. One classroom participated in the study.

#### 2.1.2. Classroom Conditions

Besides the environmental conditions around the building, the situation inside the classroom affects directly its indoor air quality. The age of the people, the ratio volume per person and the activity happening inside are important variables apart from the ventilation type.

Table 1 shows case study conditions. More details per case study can be found in Appendix A Appendix A.

### 2.2. Study Variables

#### 2.2.1. Age of the People

The age of the people in the room affects how much CO_2_ will be produced and how vulnerable they are to poor indoor air quality due to their metabolism [20,21]. Typically, children have a faster metabolism than adults [20,21], and they inhale more air per body unit [14] and, therefore, exhale more CO_2_ per body unit (weight).

Therefore, with children as a vulnerable group, an adequate environment requires lower values of CO_2_ concentrations than non-vulnerable groups.

#### 2.2.2. Ratio of Volume-People

The ratio volume-person in the classroom is a direct factor in the CO_2_ concentration in the room, therefore, also in the IAQ level. This variable represents the air volume of the classroom divided by the number of people in the room. The lower the volume per student, the higher the CO_2_ concentration per air volume and the lower IAQ assuming constant pollution sources and ventilation conditions.

#### 2.2.3. Activity Inside the Classroom

Activities that imply movement increase the breathing rate, which leads to an increase in the amount of CO_2_; thus, the type of activity happening in the room is a key factor to consider. In addition, a higher voice volume when speaking or singing causes more emissions than a normal to low voice volume [22].

#### 2.2.4. Ventilation

The most crucial variable in this study is the ventilation of the room. There are many methods to improve IAQ (filtration, forced ventilation, air purifying, etc.) but one very commonly used method is dilution, which reduces the presence of air pollutants through natural ventilation. Four natural ventilation methods have been assessed in this paper, from less to more efficient: door opening, window opening, the opening of all windows and doors, and cross window-door ventilation by allowing a crossed air flow direction.

Door opening allows an exchange of air with another interior space; this space could be an empty corridor and could have some windows open. This ventilating method is adequate when the surroundings of the schools are noisy or polluted.

Windows opening allows air exchange with the outside which is more efficient than only door opening in improving IAQ. Opening the windows is adequate when the surroundings of the school are calm and clean, and opening the door involves a disturbing and loud environment from the corridor.

The two most efficient ventilating methods are opening all the windows and doors so the air can flow through the room and opening windows and doors in a crossway through the room so the air can flow through the whole classroom space.

Other non-studied ventilation methods like forced ventilation and air purifying can be adequate solutions especially when the weather does not allow windows to open or the surroundings are noisy and or polluted.

#### 2.2.5. Sensor Setup and Monitoring Method

The only instrumentation used during the study was a sensor to characterise air quality. The sensor was a Technology Solid State Electrolyte Sensor by ARIAM. The specifications of the instrument are:Detection range: 400 to 4000 ppmPrecision: ±20%Power supply: 5 ± 0.2 VDCConsumption: 300 mWOperation range: −10 °C to 50 °C5% to 95% of relative humidity (no condensation)Pre-heat: 2 h

The data collected with this sensor was CO_2_ (parts per million), temperature (°C), humidity (%), atmospheric pressure (hPa) and particles of 0.3 µm, 0.5 µm, 5 µm, 1 µm, 2.5 µm, 4 µm and 10 µm, and the total suspended particles (TSP). However, to analyse the IAQ and study the ventilation methods, the primary variable considered was CO_2_ concentration.

This sensor was used in all the educational centres for one day. The sensor was placed in a representative place of the classroom according to an expert technician and registered CO_2_ concentrations throughout the day as well as changes in activities and in ventilation methods and their timing.

### 2.3. IAQ Modelling

The variables were not studied in a controlled space but in a real classroom; therefore, an important part of the study was visualizing and comparing the same variable in all the study cases to understand how it affects the IAQ regarding other factors.

Additionally, the study of the variables helped to determine an adequate volume per person ratio to maintain safe levels of CO_2_, the recovery times depending on the ventilation scenario, and a prediction model to anticipate a poor IAQ.

#### 2.3.1. Age of the People

In order to analyse how age affects the IAQ, classrooms with similar characteristics but different students’ ages were compared.

#### 2.3.2. Ventilation

During the study, the scenarios were defined depending on the classroom occupation and the existence of activity in it:

Set up C describes the ventilation state during the lecturing and set up R describes the ventilating method, if any, after the lecturing to recover air quality. Figure 1 shows an example of possible scenarios:C1Everything close;C2Door open;C3Window open;C4Door and window open;C5Successive classes with everything closed;C6Class with an activity involving movement happening.


R1Everything close;R2Door open;R3Window open;R4Door and window open;R5The door and window are open with cross ventilation.


In all the study cases, measurements were recorded during lecturing and recovery times; these recovery times occur during lecture breaks or during the pauses between different lectures when the classroom is empty. CO_2_ levels before lecturing, at the beginning of the day, and at the end of the sessions are also recorded.

Analysing CO_2_ evolution according to ventilation setups and classroom events it was possible to define ventilation strategies to maintain safe CO_2_ concentration levels.

#### 2.3.3. Activity inside the Classroom

Activities that imply movement increase the breathing rate, which leads to an increase in the amount of CO_2_; thus, the type of activity happening in the room is a key factor to consider. In addition, a higher voice volume when speaking or singing causes more emissions than a normal to low voice volume [22].

## 3. Results and Discussion

### 3.1. Individualized Results

Individualized results show how CO_2_ concentration varies with the studied factors studied.

Figure 2 shows the CO_2_ concentration evolution before starting the lectures, with the different events taking place, and after the lecturing day. These events show a modification in the initial variables (type of ventilation, activity, ratio of students, etc.)

It can be observed that CO_2_ concentration rises after the students enter the classroom to reach a peak of 3505 ppm in approximately three hours until a recovery time with a door opening and windows in which there is a descent of CO_2_ levels to 1000 ppm. After the comeback of students and the closing of the classroom, it rises again to 2600 ppm (one hour later). Conclusive results were achieved as the equation of the CO_2_ depending on the time for the different ventilation strategies in the recovery time or the correlation between the CO_2_ of the room and the volume ratio in the room. All classroom results can be found in Appendix A Appendix A–S11.

### 3.2. Ventilation Strategies in the Recovery Episodes

Figure 3 plots the CO_2_ concentration decrease according to different ventilation strategies during recovery times (empty classrooms) and their corresponding CO_2_ decrease equations.

Results show how cross ventilation is the fastest natural ventilation method for reducing CO_2_ concentration levels as the airflow goes through the whole room, and there is air exchange between the outside and the inside of the building.

Using an example of a classroom shown in Figure 3 and assuming 2000 ppm of CO_2_ concentration before starting the recovery time, it would take 200 min with no apparent air ventilation (classroom closed) to achieve 1000 ppm (as maximum recommended value), 47.62 min with only a door open, around 30 min with doors and windows open, and only approximately 14 min with cross ventilation.

Therefore, depending on how long the recovery time can be, one or another ventilation strategy would be optimal. For instance, if it is the end of the class and everyone is leaving the building, it could be acceptable to have a recovery time with no ventilation, but if students must come back for the next lecture, more active ventilation strategies are needed.

### 3.3. Correlation between the Evolution of CO_2_ and the Class Air Volume-Ratio

Another result of the study is the lineal equation that links the CO_2_ evolution in ppm per minute versus the classroom air volume per person ratio. This relation is for a room with no ventilation and has an R2 of 0.94.
CO_2_ evolution (ppm/min) = −3.2487 · Volume/person + 58.495(1)

With (1) it is possible to calculate the CO_2_ increase rate, measured in ppm of CO_2_ per minute, for a known volume (m^3^ of the room) and number of students. For example, a room with 150 m^3^ and 20 students would have a CO_2_ increase rate of 34 ppm/min and the same room with 13 students would have a CO_2_ increase rate of 21 ppm/min (see Appendix A Appendix A). Thus, considering an initial indoor CO_2_ concentration of 420 ppm, it would take 11.2 min to reach the 800-ppm threshold for the first example (keeping everything closed) and 18.1 min for the second example.

These results together with the CO_2_ evolution during recovery times were used to publish a COVID-19 protocol in classrooms [23] since they are effective results that can be applied in any classroom in a simple way and noticeably improve the IAQ [24].

### 3.4. Impact of the Surroundings in the IAQ

The comparison of CO_2_ concentration levels throughout the day in different schools allows us to interpret how surrounding conditions are impacting the IAQ of the room. Figure 4 shows the CO_2_ throughout the day of six different educational centres with a similar volume-student ratio (from 7 to 8.93 m^3^ per student). Nevertheless, the ventilation strategy and the age of the students were different, so they will be considered in the following discussion. The ventilation strategy per classroom is represented in supplementary materials.

In Figure 4, CO_2_ concentrations on IAQ of IES Serranía outstand the others by being noticeably higher. In this high school, the measurements were taken with students from 15 to 16 years old, and the centre is located near an open mine. This polluted outer environment leads to less frequent natural ventilation and air exchange. Moreover, this is a newly built centre with better insolation than older centres which can explain the high concentrations of CO_2_, surpassing 5000 ppm.

On the contrary, CEIP Jaume I has lower values. In this school, the measurements were taken with 20 students 9 and 10 years old. The lower values of this centre in comparison to the IES Serranía can be explained by the higher ratio volume per student in this centre (8.57 m^3^/students and 6.45 m^3^/students, respectively). Moreover, the higher frequency of ventilation and the recovery period in the middle also helped to keep lower CO_2_ concentrations.

Figure 5 shows the maximum, average and minimum values of CO_2_ of all the study cases. Since the values are represented by the study case, the same educative centre can appear multiple times. The minimum value represents the CO_2_ concentration level of the outside, which helps to understand the environment where the school is. Then, the maximum and the average values show the CO_2_ concentrations that the children are exposed to in each case.

Figure 5 shows that IES Serranía has the highest CO_2_ (ppm) registered values; it also shows a wide range of values. The frequent low ventilation and the good insolation of the building can be the cause of this dispersion.

CEIP Ciudad de Bolonia is the second centre with maximum values that can be affected by the surroundings are not polluted and some other centres, represented in the graph, are in the same neighbourhood. Comparing this centre with CEIP Jaume I, with similar characteristics, and looking at their CO_2_ throughout the day, CEIP Ciudad de Bolonia had fewer air renovation periods than Jaume I, which explains the big difference and the high CO_2_ values.

Looking at CEIP Humanista Mariner, there were three classrooms and two of them had the lowest values despite the three classes had the smallest ratio volume per student value, which was around 3 m^3^ per student for the 4 and 5-year old class and 5 m^3^ per student for the 11-years old classroom. This can be explained due to that the volume per total body unit of students in these classrooms is lower in the 11-year-old class compared to the 4- and 5-year-old classes, which supports previous experiences [17].

## 4. Conclusions

The goal of the article was to provide guidance on the design of highly efficient ventilation strategies to improve the air quality of schools’ classrooms. For this, the indoor air quality of eight educational centres was monitored during January and November 2019. It consisted of measuring the CO_2_ concentration in at least one classroom of the centre during a regular school day. These measurements were accompanied by annotations about activities and events that would affect the indoor air quality.

As a result, on the one hand, equations to calculate the recovery times for different natural ventilation strategies have been obtained, so it is possible to know the time to reach a certain CO_2_ level knowing the ventilation strategy and the previous CO_2_ concentration.

On the other hand, a relationship between the CO_2_ evolution and the volume per person in the room has been established to predict the CO_2_ concentration increase in a space with no ventilation.

Finally, the impact of the surroundings and the isolation of the building in the IAQ has been analysed. The newer buildings tend to have better isolation, which in a certain manner reduces the residual natural ventilation produced by bad isolation systems, which has to be considered when designing ventilation strategies. Moreover, pollution and noise levels of the school surroundings strongly affect the ventilation strategy selected.

The methodology employed, using carbon dioxide as the only IAQ indicator, has a downside that pollutant concentrations and sources are not considered. Factors such as the distance between the ventilation channels and the difference in the wind pressure between the inside and the outside should be considered for a more precise result. Moreover, this paper only considers natural ventilation as a ventilation strategy. In the case of highly polluted environments, it would be necessary to study other types of ventilation strategies such as forced ventilation, air purification, or a combination of both. Therefore, even though the conclusions and results of this work can be widely applied to multiple scenarios, there are many factors that deserve a deeper analysis in other cases.

Nonetheless, the tools developed allow school managers to create highly efficient ventilation strategies to maintain healthy indoor air quality by diminishing the room’s capacity, increasing the volume per ratio variable and or doing calmer and quieter activities. It also helps to.

Finally, this study has been the base for the creation of a COVID-19 protocol [23] to ensure a better IAQ in classrooms in Spain.

## Figures and Tables

**Figure 1 ijerph-20-06757-f001:**
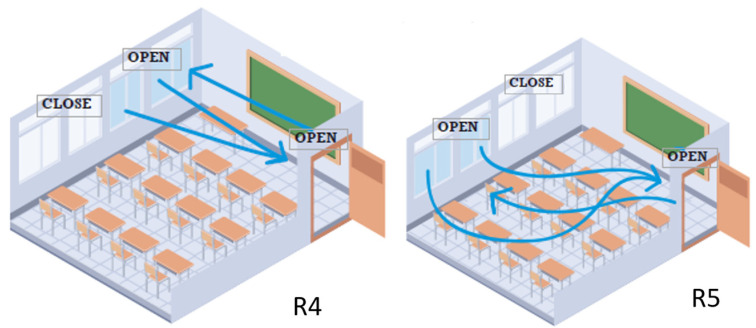
Set up R4 Door and window open and R5 cross-ventilated.

**Figure 2 ijerph-20-06757-f002:**
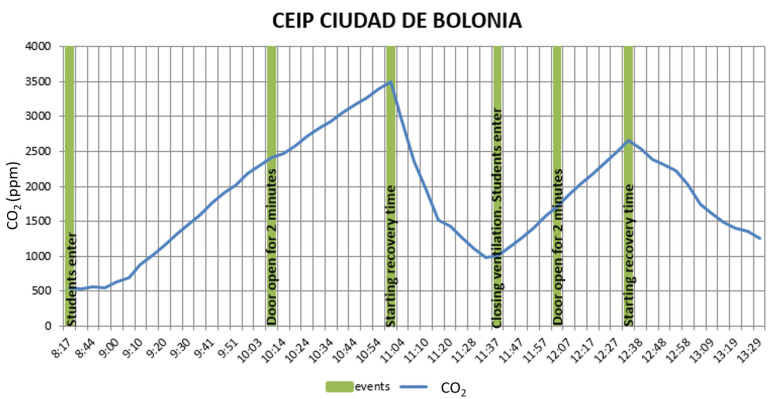
CO_2_ (ppm) and the events in CEIP Ciudad de Bolonia.

**Figure 3 ijerph-20-06757-f003:**
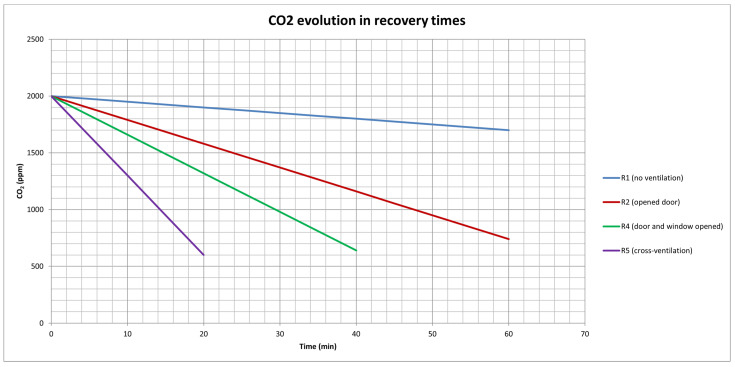
Graph showing the average CO_2_ evolution in recovery times (from the eight centres analysed) depending on the ventilation strategy used (R1–R5).

**Figure 4 ijerph-20-06757-f004:**
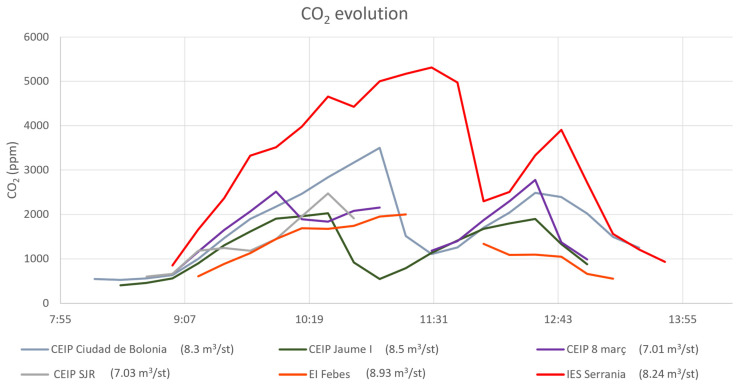
CO_2_ throughout the day in 6 educational centres with a similar volume-student ratio.

**Figure 5 ijerph-20-06757-f005:**
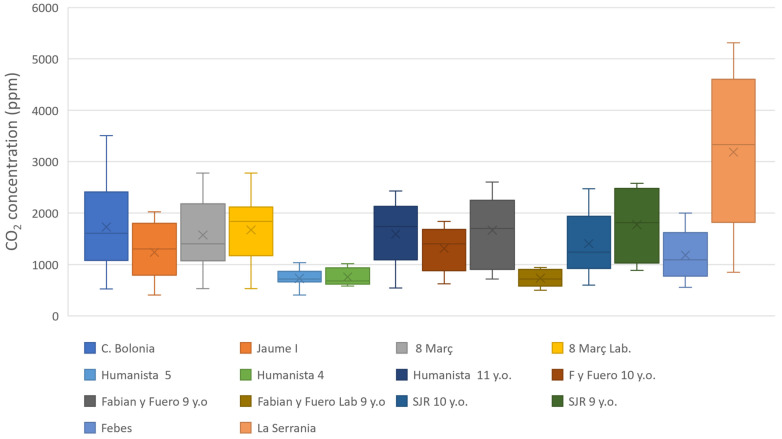
Values of CO_2_ concentration (ppm) recorded in each study case.

**Table 1 ijerph-20-06757-t001:** Description of the classrooms.

School	Students Age (Years Old)	N of Students	Classroom Volume (m^3^)	Volume Ratio (m^3^/st.)	Sensor Location (m)
CEIP Ciudad de Bolonia	9–10	19	157.7	8.3	1
CEIP Jaume I	9–10	20	171.5	8.56	1
CEIP 8 de Març	10–11	22	184.3	8.38	1
	11–12	22	311	14.14	1
CEIP HumanistaMariner	5	21	60.8	2.89	1
	4	21	60.8	2.89	1
	11	27	142.1	5.26	1
CEIP Fabián y Fuero	9–10	15	197.1	13.14	1
	8–9	17	197.1	11.59	1
	8–9	16	180.9	11.31	1
CEIP San Juan de la Ribera	9–10	15	105.4	7.02	1
	8–9	24	123.1	5.13	1
EI Febes	1–2	12	107.6	8.97	0.8
IES La Serranía	15–16	23	148.4	6.45	1

## Data Availability

Not applicable.

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
