# Peer review of "Evaluation of Different Natural Ventilation Strategies by Monitoring the Indoor Air Quality Using CO_2_ Sensors"

_ijerph, 2023, doi:10.3390/ijerph20186757_

Round 1
Reviewer 1 Report (New Reviewer)
From my point of view the topic “Evaluation of different natural ventilation strategies by moni-toring the indoor air quality using CO2 sensors.”, is interesting.
The paper is generally well structured and written. I found the idea interesting and worthwhile to review.
The thermal comfort (thermal environment in building) parameters/requirements have been described in different comfort standards. In various comfortable buildings around the world, for thermal environment air conditioners are installed. Generally, air handling units and/or ventilation systems are used to introduce fresh air in to buildings. In general, the buildings are airtight. Therefore, is not logical to use the natural ventilation in buildings.
Disadvantages of natural ventilation: irregular ventilation, air pollution, acoustic pollution, light pollution.
Therefore, please highlight the originality/innovation and significance of this study.
In the conclusion part, the deficiencies of this study or the future research directions should be added in more details. I miss the comparison to similar studies.
The article is logical and set in accordance with scientific principles.
Author Response
Dear Reviewer,
The authors thank you for your revision and enriching comments. To be able to justify the use of natural ventilation, especially in schools, I would like to put the author in the context and motivation of this research.
This research started as result of a need of deeper knowledge from public administrations about how to favour good natural ventilation in times of post COVID, when it was mandatory in Spanish schools to have leave open windows with certain frequency to improve air quality and try to avoid contagions. Moreover, in a climate change context and in an urgent need of reduction of energy use, without taking into account the energy poverty of a third of the total population of the world that cannot count with electronical devices, researching on natural ventilation strategies is more than necessary and any insight gained significant.
Moreover, regarding air conditioner systems, they do not renew the air if not that they recirculate it. Therefore, they do not constitute a solution to reduce CO2 concentrations that, as explained in the introduction, lead to reduced attendance of students in schools and increased symptoms of headache, fatigue, difficulty in concentration, etc.
For all these reasons, the authors believe that the need, originality and significance of the study is proved. Finally, we would like to point up that the results of these study were used to create a COVID-19 protocol for classrooms that has been applied in Spanish schools.
Sincerely,
Victoria Lerma
Reviewer 2 Report (New Reviewer)
The study entitles “Evaluation of different natural ventilation strategies by monitoring the indoor air quality using CO2 sensors “ by Sánchez-Fernández et al. addresses an important concern of air quality in school premises. Paper also points at the sources and the importance of CO2 concentration on overall AQI (air quality index) in any environment.
Please find my critiques and suggestions:
Methods section 2.3.4 Ventilation: Did the authors check these parameters both, with and without the students in the class? If, yes then please clarify this in the section and relative results. It stated in the manuscript that the exhaled CO2 from the subjects (students) is one of the major contributors to the total CO2 concentration in the room/rooms. Also, the study aims to determine the maximum no of students per room in order to have the CO2 levels at normal concentrations. Therefore, it will be very important to know the concentrations of CO2 in the rooms without any students and an comparative analysis of CO2 concentrations with and without the students in those rooms will answer the research question better.
Minor grammar check and editing required
Author Response
Dear Reviewer,
First of all, many thanks for your comments and suggestions. I will try to answer and clarify for questions.
We have measure parameters with and without students. These measurements without students were registered during the recovery times (the period of time with no people in the room) and took place during lecture breaks, during the pause times between lectures and before the start of lectures in the morning and at the end. During all those times, we observed a descend on CO2 levels that equalled outdoors CO2 levels when recovery times where longer (before and after the lectures). These values can be observed in the details of CO2 events per centre plotted in Supplementary materials (S2-S12).
Please, see the modifications made on the second paragraph of 2.3.4 bullet point and on second paragraph of 3.1.
Yours sincerelly,
Victoria Lerma
Reviewer 3 Report (New Reviewer)
- Authors have chosen an interesting and valuable topic to investigate. However, the study design, the understanding of the authors, and the presentation of the article leaves a lot to be desired. I cannot recommend the publication of this work. Below are some points where I believe the authors definitely need to improve. To clarify though, this is not an exhaustive list and authors need to put a lot more work into this kind of papers.
- The language usage needs to be improved. In several parts of the text, due to use of improper structure and grammar, content comprehension is being affected.
- The Abstract is heavy on qualitative descriptors. It would be useful to have a couple of lines relating to some concrete findings. Again, the quality of English here means I am not even sure what the authors are trying to say.
- Introduction - there are other manners of improving air quality, in addition to dilution.
- CO2 can relate well with pollutant concentrations that have a human origin.
- Authors need to clearly demarcate that there are effects of exposure to high cocnentrations of CO2 but, for most of the studies discussed, the consequences are from poor ventilation, which CO2 is a marker for. As the authors mention, CO2 can be used to measure ventilation. More often than not, when a certain symptom or well-being related impact is associated with high CO2, it is about bad ventilation and not the impact of just CO2, in itself.
- Current versions of ASHRAE standard do not refer to a CO2 limit
- Methods - children may exhale more CO2 per body weight, but that does not mean they exhale more CO2 than adults in absolute terms. There is of course the additional factor that children in a classroom are far more likely to be engaged in a higher activity rate than adults in an office. I am not convinced by the argument in Section 2.2.1
- Lower volume per person does not necessarily mean poor IAQ. If it were so, warehouses, with high ceilings, would never have poor IAQ.
- +/- 20% precision, for these kind of measurements is too much variation
- Authors could have provided measurement locations on classroom floor plans, possibly even in supplementary materials.
- Results - Fig 3. There would not be a situation with no/0 ventilation. there is always some kind of infiltration going on.
- What is the point of Equation 1? Just because you can form linear regression equations, does not mean you should. Is there a physics of the situation that says this kind of relationships can be reliably used under different circumstances? Figure 4 itself implies that such a relation has little to no value.
- What does "better insolation" have to do with ventilation?
Extensive editing of English required. In its present state, there are large stretches of the article which make no sense or are just confusing.
Author Response
Dear Reviewer,
Many thanks for your enriching comments and suggestions. I will try to answer and clarify for questions.
First of all, I would like to apologise for the English level, I have tried to improve it. Regarding the rest of the comments, I would like to deal with them point by point:
- The Abstract is heavy on qualitative descriptors. It would be useful to have a couple of lines relating to some concrete findings. Again, the quality of English here means I am not even sure what the authors are trying to say.
The Abstract has been modified.
- Introduction - there are other manners of improving air quality, in addition to dilution.
This have been clarified in the Introduction
- CO2 can relate well with pollutant concentrations that have a human origin.
Right
- Authors need to clearly demarcate that there are effects of exposure to high cocnentrations of CO2 but, for most of the studies discussed, the consequences are from poor ventilation, which CO2 is a marker for. As the authors mention, CO2 can be used to measure ventilation. More often than not, when a certain symptom or well-being related impact is associated with high CO2, it is about bad ventilation and not the impact of just CO2, in itself.
Noted
- Current versions of ASHRAE standard do not refer to a CO2 limit
Noted
- Methods - children may exhale more CO2 per body weight, but that does not mean they exhale more CO2 than adults in absolute terms. There is of course the additional factor that children in a classroom are far more likely to be engaged in a higher activity rate than adults in an office. I am not convinced by the argument in Section 2.2.1
This has been properly clarified.
- Lower volume per person does not necessarily mean poor IAQ. If it were so, warehouses, with high ceilings, would never have poor IAQ.
We do not agree with the comment. Lower ratios of air volume per person always means lower IAQ than higher ratios with the same ventilation conditions, of course, considering same conditions and pollution sources.
- +/- 20% precision, for these kind of measurements is too much variation
This precision is obtained from the sensor catalogue. Please, check it in https://docplayer.es/46565189-Sistemas-de-telegestion.html
- Authors could have provided measurement locations on classroom floor plans, possibly even in supplementary materials.
Yes, but it is not possible at this moment.
- Results - Fig 3. There would not be a situation with no/0 ventilation. there is always some kind of infiltration going on.
We are aware of it and because of that we refer to building isolation conditions in our fourth section. But in Scenario 1, with no ventilation, we are referring to that there is any open or window opened. We added a remark on that in the text.
- What is the point of Equation 1? Just because you can form linear regression equations, does not mean you should. Is there a physics of the situation that says this kind of relationships can be reliably used under different circumstances? Figure 4 itself implies that such a relation has little to no value.
We sorry, but we don’t understand this comment. The equation refers to the increase of CO2 concentration per minute according to the number of students in a certain classroom of a given volume when it is closed and theoretically there is no ventilation. On the contrary, figure 4 is representing a series of events during a school day time in a classroom and reflects how these events affect CO2 concentration levels. Among these events, break times and different types of ventilations are included. Please, check Supplementary material information (S2-S13) to observe the events and understand their influence.
- What does "better insolation" have to do with ventilation?
In Spain and generally in Mediterranean basin countries, it is frequent to find not good isolated buildings that allow certain circulation of air due to bad isolating systems. Therefore, on the contrary, a good isolation means that this residual ventilation do not take place. This effect, that contribute to a better energy efficiency of the building, highlights the need of active ventilation strategies to maintain IAQ levels.
Yours sincerely,
Victoria Lerma
Round 2
Reviewer 2 Report (New Reviewer)
None. Comments addressed well.
Author Response
Dear Reviewer,
The authors want to thank you very much for all your dedication and comments provided,
Very kind regards,
Victoria
Reviewer 3 Report (New Reviewer)
I do not feel authors have made any majo changes. The list of items I had mentioned, and as I had also menioned in the forward to it, was just an indicative list. Authors have not made an effort to improve the language comprehendibility or the presentation. There is no change in the mehodology and the reporting and discussions style. This work doe snot merit publication.
English needs to be drastically improved to reach a publciation standard.
Author Response
Reviewer 3,
We did all the "indicative" changes you asked for in your previous comments. The tone was still more unpolite this time and we, as authors who have worked during a long time in preparing the case study, implementing them, analysing results and writting this articles do not deserve this treatment. I have written to the Editors to inform them and I wanted also to inform you directly.
Nevertheless, given our honour and professionality and due to that we are under a scientifically review process, we have considered the review of the entire paper to improve the language comprehensibility and presentation. I hope this fulfill your expectations and I ask you for respect for the work of other scientists in my name and for the other papers you may receive for revision.
This manuscript is a resubmission of an earlier submission. The following is a list of the peer review reports and author responses from that submission.
Round 1
Reviewer 1 Report
We are appreciate to authors for submitting your valuable research results to the journal.
My review as follows;
This paper focuses on ‘ventilation’, there are several ventilation types as you know. I think the title should include ‘natural ventilation’.
I think that the geometric form, dimensions, location of windows and doors of the eight classrooms, which are the subject of this study, should be presented along with schematic drawings.
Also, location and number of CO2 sensors should be provided.
It is difficult to understand that only two results of Figure 2 and Figure 3 are presented when the subject of measurement is 8 classrooms.
I don't know if Figure 4 is the result of one classroom or the average of eight classrooms.
The text on lines 265-267 is thought to be an explanation for fig 4.
I think Figure 5 is the main results and suggestion of this paper. But I think Fig.5 have some limitations. As you know, the main factors that affect natural ventilation are the size of openings, the distance between openings (height difference), and the wind pressure difference between indoors and outdoors. Figure 5 does not take them into account and simply deals with the volume of the room.
Indoor and outdoor wind pressure is a very important factor influencing the amount of natural ventilation. I don't understand why the wind pressure was not considered in this study even though the measuring device can measure the wind pressure.
I think that this paper does not draw meaningful conclusions due to insufficient preparation for measurement and physical background knowledge about natural ventilation. Ventilation performance evaluation using CO2 has recently been a lot of research, so I hope that the authors refer to this to derive meaningful research in the future.
Reviewer 2 Report
In general, this work has very limited contribution to the scientific community. It has scientific flaws.
Although, English is not my mother tongue, the language in this work has to be improved a lot!!!
Plagiarism can be found in the manuscript, just in the first paragraph.
The authors do not understand the terms and the theory of the IAQ.
For example:
Reference such as this
Morse, C. “How the respiratory system affects metabolism” in Better Breathing by PowerLung. Available online: 459 https://blog.powerlung.com/better-breathing/how-the-respiratory-system-affects-metabolism (accessed on 29 November 2022).
has not to be used, since numerous other studies exist on this issue in the academic literature.
Line 8:
‘Indoor air quality depends on air pollutants inside a building, and ventilation improves it’
This not always true. You could replace the word ‘’improves’’ with ‘’affects’’.
Line 10: the CO2 concentrations are an indicator of the human presence and certainly not an accurate indicator of IAQ.
Line 65: Even in schools, the CO2 is not the only pollutant that affects IAQ. The PM, the VOCs pose a higher threat to human health. It is not correct to relate IAQ with only the CO2 concentrations. Furthermore this is not true for shopping centres or hotels.
Line 70: ‘whether there is any kind of ventilation’ perhaps the author wanted to say ‘no mechanical ventilation’. Any room has a measurable air exchange rate, at least due to unintended air infiltration and exfiltration.
Line 286 ‘’increase velocity’’ Please… this is not science.
Please see the meaning of thermal comfort
https://en.wikipedia.org/wiki/Thermal_comfort
The results, did not reveal nothing new and nothing useful. Experimental work is always valuable, but a good interpretation of the results must follow. If these were data treated and presented in another way, probably they could lead to mush more interesting interpretation which will be an adding value for the scientific community.
